# Health Impacts of the COVID-19 Lockdown Measure in a Low Socio-Economic Setting: A Cross-Sectional Study on Reunion Island

**DOI:** 10.3390/ijerph192113932

**Published:** 2022-10-26

**Authors:** Adrian Fianu, Hind Aissaoui, Nadège Naty, Victorine Lenclume, Anne-Françoise Casimir, Emmanuel Chirpaz, Olivier Maillard, Michel Spodenkiewicz, Nicolas Bouscaren, Michelle Kelly-Irving, Emmanuelle Rachou, Cyrille Delpierre, Patrick Gérardin

**Affiliations:** 1Institut National de la Santé et de la Recherche Médicale (INSERM) Centre d’Investigation Clinique CIC1410, Centre Hospitalier Universitaire de La Réunion, CEDEX, 97448 Saint-Pierre, France; 2Center for Epidemiology and Research in POPulation Health (CERPOP), Université de Toulouse, Institut National de la Santé et de la Recherche Médicale (INSERM), Université Paul Sabatier (UPS), 31000 Toulouse, France; 3Unité de Soutien Méthodologique, Centre Hospitalier Universitaire de La Réunion, 97400 Saint-Denis, France; 4Moods Team, Centre de Recherche en Épidémiologie et Santé des Populations UMR-1178, 94275 Le Kremlin-Bicetre, France; 5Observatoire Régional de la Santé—La Réunion, 97400 Saint-Denis, France

**Keywords:** COVID-19 pandemic, lockdown, Reunion Island, living conditions, self-reported health, social inequalities in health, social epidemiology

## Abstract

In March 2020, the French government implemented nation-wide measures to reduce social contact and slow the progression of the emerging coronavirus responsible for COVID-19, the most significant being a complete home lockdown that lasted 8 weeks. Reunion Island is a French overseas department marked by large social inequalities. We draw the hypothesis that distancing and lockdown measures may have contributed to an increase in the social inequalities in health (SIH) on Reunion Island. The aim of our study was to describe the SIH during lockdown in the Reunionese population. We implemented a cross-sectional telephone survey conducted between 13 May and 22 July 2020, using a retrospective data collection on the lockdown period. A total of 892 adult participants (≥18 years) were recruited in the 114 large Reunionese neighborhoods using the quota method within the national “White Pages” telephone directory. Degraded psychological states, an increase in addictive behaviors, difficulties in accessing food, a decrease in physical activity, delayed medical appointments, violence against women, and health problems in children were driven by the socio-economic characteristics of the population, most often to the disadvantage of social groups exposed to poor living conditions. These results suggest that the COVID-19 lockdown contributed to an increase in SIH.

## 1. Introduction

On 17 November 2019, a novel coronavirus designated as Severe Acute Respiratory Syndrome Coronavirus 2 (SARS-CoV-2) responsible for COVID-19 became an epidemic in Wuhan, Hubei province, China. By 11 March 2020, SARS-CoV-2 became a pandemic [1]. In the following days, the French government implemented nation-wide measures to reduce social contact and slow the progression of this new emerging pathogen, the most restrictive being social and physical distancing and a complete home lockdown that lasted 8 weeks from 17 March to 11 May. These universal prevention measures (i.e., targeting the overall general population) were aimed at flattening the epidemic curve to prevent the healthcare service from being overwhelmed [2]. However, these radical preventive measures may have worsened vulnerabilities in the general population. For example, collateral damage such as psychological distress affected more isolated [3] or deprived persons of the community [4], especially the elderly people [5] and those experiencing poor living conditions [6].

To better understand these adverse effects on the population, it is important to address the wider issues surrounding the public health impact of the COVID-19 pandemic on both geographic inequalities in health, particularly in low socio-economic settings, and social inequalities in health (SIH) [7]. SIH are socially constructed, unjust, and avoidable disparities in health [8]. They are observed in variations in health states that follow a socio-economic gradient across the population. Thus, people at the top of the social gradient are the healthiest, with health states gradually worsening as one goes down this gradient all the way to the most precarious people at the bottom.

Reunion Island is a French overseas department of 860,000 inhabitants located in the South-West Indian Ocean (SWIO) region, with a long history of social inequality [9]. According to the national institute of statistics and economic studies (Insee) dedicated to the collection, analysis, and dissemination of information on the French economy and society (https://www.insee.fr/en/accueil (accessed on 9 October 2022)), this territory still faces a huge level of economic insecurity, with 40% of its population living below the mainland poverty threshold [10]. Economic difficulties, family structure, and housing are the main social factors that differentiate the Reunionese neighborhoods from one another [10]. Moreover, even in the better-off neighborhoods, the poverty rate remains much higher than in mainland France [10]. Regarding the situation before the COVID-19 pandemic, Reunion Island’s poverty rate was among the three highest of all 101 French regional estimates [11]. This statistic testifies to the social vulnerability and poor living conditions that many inhabitants of Reunion Island have had to face in their daily lives for many years.

Previous work has shown that preventive measures to tackle the SARS-CoV-2 epidemic may have had a deleterious effect on health, particularly mental health, in less-advantaged populations [3,12]. In the context of social vulnerability observed on Reunion Island, we hypothesized that distancing and lockdown may have contributed to the increase in SIH on Reunion Island, in terms of stress, anxiety, addictions, violence, physical inactivity, difficulties in healthcare access, and health problems in children.

Our reflections are laid out in the following conceptual model (Figure 1):

In this conceptual model, which is an overall representation of the nested structure of health determinants [13] under a socio-ecological perspective [14], we assumed a cumulative effect: in the pandemic context with an eight-week long home lockdown to curtail the SARS-CoV-2 epidemic, the more unfavorable the living conditions (in terms of neighborhood deprivation, poor housing, low individual socio-economic position, and epidemic exposure), the higher the probability of observing degraded behaviors or health states (e.g., stress, addictive behaviors, violence suffered, and sedentary lifestyles).

The aim of our study was to describe the SIH during lockdown in the Reunionese population conditional on neighborhood deprivation, housing conditions during the lockdown, and the individual socio-economic position.

## 2. Materials and Methods

### 2.1. Research Design

The Ré-Conf-ISS (Réunion-Confinement-Inégalités Sociales de Santé) one-time cross-sectional telephone survey was conducted just after the lockdown, between 13 May and 22 July 2020.

### 2.2. Study Population and Sampling Method

The participant eligibility criteria for inclusion were the following: aged 18 years and over, living in Reunion Island before the beginning of the lockdown, spending the entire lockdown (8 weeks) on Reunion Island, and being able to answer the questionnaire (without assistance). 

Participants were sampled within the national “White Pages” telephone directory (https://www.pagesjaunes.fr/pagesblanches (accessed on 9 October 2022)) using the quota method. Quotas were defined by the proportion of adult residents in each large neighborhood [10]. According to the Insee, Reunion Island is composed of 114 large neighborhoods, which comprise a number of adult inhabitants (≥18 years) per large neighborhood spanning from 1419 to 14,262 [10]. For a given large neighborhood, each of the seven trained researchers of the Reunion Island Clinical Investigation Center (CIC) survey platform had a number of participant inclusions to reach (i.e., quota). The inclusion process involved the following steps: (1) in a given municipality, an arbitrary choice by the researcher of one person per household from the national “White Pages” directory was contacted by telephone; if the contact was successful, (2) information on the study was provided; (3) eligibility criteria were checked; (4) the home address where the person was confined was localized; and (5) informed oral consent was collected and the quota achievement was checked.

### 2.3. Data Collection

Telephone interviews were conducted by the CIC survey platform using a 115-item retrospective questionnaire. This questionnaire covered the lockdown period in the following domains: localization (at the large neighborhood level) and type of housing, housing equipment, environment in the vicinity of the home, individual socio-economic position, exposure to the SARS-CoV-2 epidemic, mental health, addictive behaviors, nutrition, and healthcare use. Two specific domains of the questionnaire gathered information on violence against women and children’s health based on parent’s statements.

### 2.4. Exposure Factors

#### 2.4.1. Large Neighborhood Deprivation Level

Given the Reunionese large neighborhood deprivation typology [10], the 114 Reunionese large neighborhoods are classified into five homogenous groups based on their level of deprivation: Group 1 was composed of “urban neighborhoods facing multiple socio-economic difficulties” (G1 = 11.3% of the adult population); Group 2 was composed of “predominantly rural neighborhoods inhabited by poor homeowners” (G2 = 22.6%); Group 3 was composed of “vulnerable neighborhoods located close to city centers” (G3 = 19.4%); Group 4 consisted of “less poor neighborhoods located far from city centers” (G4 = 29.4%); and Group 5 consisted of “better off neighborhoods” (G5 = 17.3%) [10]. These five ordinal groups describe a socio-economic gradient across the population.

#### 2.4.2. Housing Conditions during the Lockdown

Five binary indicators (yes/no) of housing conditions during the lockdown based on ad hoc definitions were: living in overcrowded conditions, a potentially high mental load, daily annoyances in the home, optimal access to the outside, and poor Internet access. Living in overcrowded conditions corresponded to a ratio < 1 between the number of rooms in a home (numerator) and the number of individuals living in that home during the lockdown (denominator), given that two children under 10 years of age could share the same room without this constituting overcrowding. A potentially high mental load was defined by sharing a home with ≥2 children (<18 years) during the lockdown. Daily annoyances in the home consisted of the following: too many inhabitants according to the participant’s self-assessment, humidity problems, noise problems, and temperature problems (too hot or too cold). Optimal access to the outside was defined by the following combination of accessibility situations: access to a garden or kour and access to a balcony, terrace, or veranda, and access to a public park, garden, or play area in the immediate vicinity of the home.

#### 2.4.3. Individual Socio-Economic Position

Six indicators of individual socio-economic position were considered: usually living alone (yes/no), university degree as education level (yes/no), socio-professional category (employed/unemployed/retirees), experience of financial difficulties related to the COVID-19 health crisis (yes/no or don’t know), housing occupancy status (homeowners/renters/rent-free), complementary health insurance status (positive, i.e., benefiting from a private complementary health insurance/negative, i.e., lacking complementary health insurance or benefiting from the free public complementary health insurance).

#### 2.4.4. SARS-CoV-2 Epidemic Exposure

We relied on self-reported infection (yes with PCR test confirmation/no/don’t know) as the indicator of SARS-CoV-2 epidemic direct exposure. We considered the knowledge of an infected close relative living on Reunion Island or elsewhere (yes/no/don’t know) as an indicator of indirect exposure to the SARS-CoV-2 epidemic being able to degrade the psychological state.

### 2.5. Demographic Factors and Baseline Health State

Demographic factors included sex (woman/man) and age. The baseline health state included a history of chronic disease or a health problem requiring regular visits to the doctor (yes/no/don’t know). These factors were used as a proxy of the baseline (i.e., prior to the COVID-19 pandemic) health-risk level, which could differ between the social groups to be analyzed in the multivariable models (see statistical analysis).

### 2.6. Endpoints

Some of the most characteristic behaviors and individual health states related to the lockdown period were taken as outcome variables or endpoints. 

#### 2.6.1. Primary Endpoint

The primary endpoint was a high level of stress during the lockdown. For the purpose of a telephone survey, we transposed a Likert visual analog scale used for a web-based inquiry to rate the perceived stress during the lockdown. A high level of stress during the lockdown was defined by a score > 6 on this scale ranging from 0 (no stress) to 10 (maximum imaginable stress) [15]. 

#### 2.6.2. Secondary Endpoints

The secondary endpoints included other mental health indicators (i.e., deteriorated psychological states and positive appreciation of the lockdown), addictive behaviors, nutrition (i.e., difficulties in accessing food and decreases in physical activity), healthcare use (i.e., delayed medical appointments), violence against women, and children’s health.

A deteriorated psychological state during the lockdown was a binary endpoint defined by the presence of at least one of the following five situations: currently not feeling calm in daily life and/or experiencing lockdown-related worsening of a psychological condition pre-existing the epidemic and/or a psychological condition triggered by the pandemic or lockdown and/or a perception of one’s social relationships as very bad or somewhat bad and/or worries about ones’ job or work situation in the immediate future.The lockdown positive appreciation score was a count endpoint defined as the sum of behaviors or attitudes for having engaged in at least one of the following thirteen relaxing activities: refocusing, enjoying the quiet, breathing less polluted air, having conversations, reading, writing, watching movies/series, playing games, cooking, DIY (do it yourself), gardening, doing nothing, and doing something else that feels good. The higher the score, the better the positive appreciation of the lockdown. A zero (minimum) value meant no positive appreciation of the lockdown or unknown level of appreciation.The increase in addictive behaviors during the lockdown was a binary endpoint. It featured the use of at least one of the following items: screens, alcohol, tobacco (cigarette/e-cigarette), or psychotropic drugs (cannabis/zamal/chimique, i.e., natural or synthetic cannabinoid-based drugs), and participation in virtual happy hours.The score of difficulties in accessing food during the lockdown was a count endpoint defined as the sum of difficulties for the following items: fresh fruits and vegetables, garlic/onions, eggs, flour, and other essential food items. The higher the score, the higher the number of difficulties. A zero (minimum) value meant no difficulty.The decrease in physical activity during the lockdown was a binary endpoint defined by the presence of at least one of the following three situations: sports practice at a club or an association before the lockdown and/or use of a neighborhood sports facility before the lockdown and/or perceived decrease in physical activity during the lockdown.Delayed medical appointment during the lockdown was considered as two binary endpoints sharing the same reference category. Firstly, “At the request of the medical secretariat” versus “No delayed medical appointment at all or medical appointment delayed for another reason than at the request of the medical secretariat and on its own initiative” (reference category); secondly, “On its own initiative” versus the same reference category as above.Violence against women was a binary endpoint defined by nine situations of economic or psychological abuse linked to the lockdown and the COVID-19 pandemic: using the pandemic as an excuse to increase control over family finances; depriving the female partner and the children of food, medicine, and/or hydro-alcoholic gel; issuing threats, preventing the female partner and the children from seeing a doctor when they present symptoms, or hiding their health cards; controlling and criticizing the female partner; blaming the children’s behavior on the female partner; isolating the female partner and the children by preventing or monitoring phone calls, emails, access to online social networks, etc.; forcing the female partner and the children to stay in certain areas of the home (bedroom, garage, etc.); after a separation, getting the female partner to come back to the home or returning to live in the home of the female partner; perpetrating verbal or physical abuse by blaming the female partner’s behavior on the lockdown.Health problems in children (<18 years) during the lockdown concerned the entire sibling group and consisted of the following: worry, anxiety, stress, sleep and eating disorders, concentration or attention problems, and learning difficulties. They were identified based on the statements by parents confined with their child(ren) during the lockdown and summarized in a sum score. The higher the score, the lower the health of children. At least one health problem in children was a binary endpoint based on the child health score ≥ 1 (Yes/No).

### 2.7. Statistical Analysis

A sample-size calculation is shown in Appendix B. To fulfill the study objective, we measured adjusted associations between exposure factors and each endpoint using regression models. A logistic regression model for binary outcome was used to estimate adjusted odds ratios (aORs) with their 95% confidence interval (CI). Two types of count data regression models were used to estimate adjusted incidence rate ratios (aIRRs) with a 95% CI: a negative binomial (NB) regression for modeling the child health score with an offset (equal to the logarithm of the size of entire sibling group) to adjust the at-risk population size; and a zero-inflated negative binomial (ZINB) regression for modeling the lockdown positive appreciation score and the food accessibility score, respectively. In the study, on average 7.8 participants per large neighborhood (min–max = 2–21) were selected using the quota method. Thus, to account for the non-independence of observations within the cluster defined by a large neighborhood (114 clusters), we performed all regression models using the generalized estimating equation (GEE) approach with an exchangeable (compound symmetry) working correlation matrix and a robust variance estimator [16], except for ZINB regression models wherein the variance and 95% CI were estimated using a cluster bootstrap technique with 200 re-samples [17]. 

Nested multivariable regression models were run given the theoretical framework of health determinants [13] under a socio-ecological perspective [14]. Independent variables were selected according to the study’s conceptual model (Figure 1), data quality, and the associations that were significant in bivariate analysis (at *p* ≤ 0.05). A summary of independent variables in nested multivariable regression models is shown in Appendix C. The first multivariable model (M1) included the sex, age, and history of chronic disease or health problem requiring regular visits to the doctor. We then added indicator(s) of individual socio-economic position in a second multivariable model (M2). In M3, we added to M2 indicator(s) of collective housing conditions during the lockdown. In the full adjusted model M4, we added to M3 the large neighborhood deprivation level as an upstream health determinant [14]. All multivariable models (M1, M2, M3, and M4) were computed on available data providing the same sample size. The increase in the large-neighborhood deprivation level was analyzed continuously to test a linear gradient. We assumed that housing conditions during the lockdown could interfere with the large-neighborhood deprivation level on the individual health state. Therefore, the first-order interaction term between the indicator of housing conditions during the lockdown and the large neighborhood deprivation level was tested and M4 was stratified by housing conditions when appropriate. Observations with missing data were excluded under the missing completely at random assumption. All the statistical analyses were performed using the R 4.0.3 software. Statistical significance level was set to 5%. All tests were two-tailed.

### 2.8. Ethical Considerations

The Ré-Conf-ISS study was conducted in accordance with the Declaration of Helsinki and the French law of bioethics. It was approved during the lockdown, on 30 April 2020, by the Conseil Scientifique Restreint COVID-19 of the Centre Hospitalier Universitaire de La Réunion (CHU), and registered in the public directory of the Institut National des Données de Santé (INDS: MR 5219300420) and the register of processing activities of the CHU’s Data Protection Officer (DPO) in accordance with the European regulation (Règlement Général sur la Protection des Données—RGPD). For this type of study (classified as: Recherche N’Impliquant pas la Personne Humaine—RNIPH), the regulations do not require evaluation by an ethics committee. The compliance of the Ré-Conf-ISS project with the MR-004 of the Commission Nationale Informatique et Libertés (CNIL) takes into consideration ethical aspects such as the relevance of the data collected with regard to the objectives of the research. 

Participants gave their informed oral consent to participate in the study at the beginning of the interview. They were also informed of the possibility of accessing their data and dropping out of the research by visiting the CHU’s website and finding the principal researcher contact to make such a request.

## 3. Results

The participant selection process is shown in Appendix D. The Ré-Conf-ISS study participation rate was 27% (892/3301). Among the 892 participants retained for analysis (Appendix D), 571 were women and 354 were parents confined with their child (or children, if any) during the lockdown.

### 3.1. Population Description

Table 1 shows the exposure factors, demographic factors, and baseline health state of the 892 participants, compared to regional references when available. 

The distribution of the large-neighborhood deprivation level of the five groups was similar in the study and the Reunion Island adult population (Table 1). The prevalence of poor housing conditions during the lockdown was higher than 20%, regardless of the indicator. Almost one-third of all participants (31.4%) stated having experienced financial difficulties related to the health crisis. More than one-half (55.6%) were homeowners. Of note, 78.3% of respondents lived in a single-family house, less than 10% in an apartment within a private residence, and 12.3% in social housing (data not shown in Table 1). Two participants (0.2%) reported a SARS-CoV-2 infection that was confirmed with a PCR test (Table 1). Five percent of participants (5.4%) knew that SARS-CoV-2 had infected a close relative living on Reunion Island or elsewhere. 

Table 2 completes the description of the study population and reports the study endpoints.

Regarding degraded psychological state (Table 2), 20.2% reported a high level of stress and 47.3% a deteriorated psychological state. Moreover, a large majority of participants reported an absence of lockdown positive appreciation (54.7%) and an increase in at least one addictive behavior (57.8%). The same was true for difficulty in accessing food (56.4%) and a decrease in physical activity (62.0%). Medical appointments, an indicator of healthcare use, were postponed in 21.5% at the request of the medical secretariat. Regarding specific subpopulations, 3.9% of women reported having suffered at least one situation of economic or psychological abuse linked to the lockdown and the COVID-19 pandemic, while more than two-thirds of parents (68.4%) noticed at least one health problem in their child(ren) during the lockdown.

### 3.2. Main Results: Factors Independently Associated with Behaviors and Individual Health States Related to the COVID-19 Lockdown Period

Table 3, Table 4, Table 5 and Table 6 (overall adult population) and Table 7 (women and children population, respectively) present the exposure factors independently associated with endpoints in the fully adjusted multivariable models.

#### 3.2.1. Mental Health and Addictive Behaviors Endpoints

Regarding mental health, during the COVID-19 lockdown, the factors independently associated with a high level of stress in adults (*n* = 880) were (Table 3) sex, age, experience of financial difficulties related to the COVID-19 health crisis, and history of chronic disease. On the one hand, men and older participants (60–74 years) were less likely to report a high level of stress (compared to women and 18–59 years, respectively). On the other hand, the event of a high level of stress was higher among participants experiencing financial difficulties related to the health crisis or having a history of chronic disease (compared to their respective counterparts). These associations were already significant in the first nested multivariable regression models (see Appendix A). 

Living in overcrowded conditions during the lockdown (Yes/No) was identified as an effect modifier in the relationship between the large-neighborhood deprivation level and a deteriorated psychological state (*p* = 0.039). In view of this, the fully adjusted multivariable analysis was stratified by this factor (Table 3). For adults living in overcrowded conditions during the lockdown (*n* = 275), the factors independently associated with a deteriorated psychological state were experience of financial difficulties related to the COVID-19 health crisis and history of chronic disease. Furthermore, adults experiencing financial difficulties related to the health crisis and those with a history of chronic disease (compared to their respective counterparts) were more likely to exhibit a deteriorated psychological state. For adults not living in overcrowded conditions during the lockdown (*n* = 577), the factors independently associated with a deteriorated psychological state were age, experience of financial difficulties related to the COVID-19 health crisis, and large-neighborhood deprivation level. Moreover, elder participants (60–74 years of age) were more resilient than young and middle-aged adults (18–59 years of age) as suggested by protective odds for deteriorated psychological state (same observation for the 75–90 years age group in M1 became non-significant in M2; see Appendix A), as were those that did not experience financial difficulties related to the health crisis compared to their counterparts. Finally, the probability of a deteriorated psychological state increased as the large-neighborhood deprivation level went up. In other words, inhabitants from the most deprived large neighborhoods were more likely concerned by a deteriorated psychological state than the others (Table 3). 

After multiple adjustments (*n* = 850), inhabitants of the most deprived neighborhoods were more likely to have an unfavorable assessment of lockdown than others (Table 4). However, they were less concerned by an increase in at least one addictive behavior during the lockdown (Table 4). Other independent factors associated with the addictive behavior endpoint were age, education level, and housing occupancy status. On the one hand, the likelihood of an increase in at least one addictive behavior was lower in the 60–74 years and 75–90 years age groups, respectively (compared to 18–39 years of age). On the other hand, the likelihood of an increase in at least one addictive behavior was higher for participants with a university degree level and renters, respectively (compared to their counterparts).

#### 3.2.2. Nutrition Endpoints

Regarding nutrition, the experience of financial difficulties related to the COVID-19 health crisis significantly increased the number of difficulties in accessing food during the lockdown, when controlling for other socio-economic factors (Table 5). In contrast, older participants (75–90 years of age) and those without a potentially high mental load were more likely to report no difficulty in accessing food. 

Having optimal access to the outside during the lockdown (Yes/No) was identified as an effect modifier in the relationship between the large-neighborhood deprivation level and decrease in physical activity (*p* = 0.036). In view of this, the fully adjusted multivariable analysis was stratified by this factor (Table 5). For participants with optimal access to the outside during the lockdown (*n* = 214), the only factor independently associated with a decrease in physical activity was the large neighborhood deprivation level. More precisely, the probability of a decrease in physical activity during the lockdown decreased as the large neighborhood deprivation level went up. In other words, inhabitants from the less deprived large neighborhoods were prone to diminish their physical activity during the lockdown than the others. For participants without optimal access to the outside during the lockdown (*n* = 651), the only factor independently associated with a decrease in physical activity was the education level, as shown by a higher reduction rate of physical activity among participants with a university degree (Table 5). 

#### 3.2.3. Health Care Use Endpoints

Regarding healthcare use (Table 6), three profiles increased the likelihood of having a postponed medical appointment at the request of the medical secretariat (*n* = 712): having a university degree, benefiting from private complementary health insurance, and having a history of chronic disease or health problem requiring regular visits to the doctor. Factors independently associated with a delayed medical appointment on its own initiative are presented in Appendix A.

#### 3.2.4. Economic or Psychological Violence against Women

Regarding economic or psychological violence against women (Table 7), the factors that increased the likelihood to suffer at least one situation of violence (all concerned women aged 18–74 years, *n* = 571), were: daily annoyances in the home, a potentially high mental load, unemployed status, and a history of chronic disease. 

#### 3.2.5. Health Problems in Children

According to the statement of 354 parents confined with their child(ren) during the lockdown (Table 7), the number of health problems experienced by children decreased when the study participant was a father rather than a mother (same effect in M1, M2, and M3, respectively; see Appendix A) and increased with the presence of at least one annoyance in the home (versus no annoyances). In the same way, the occurrence of at least one health problem in children was increased by experiencing financial difficulties related to the health crisis (Table 7).

## 4. Discussion

Our study, examining the relationship between socio-economic factors and a wide range of health and wellbeing measures among La Réunion residents in spring 2020, underscores the existence of SIH during the early phase of the pandemic, when the COVID-19 virus was not yet widespread on the island. Indeed, a degraded psychological state, an increase in addictive behaviors, difficulties in accessing food, a decrease in physical activity, delayed medical appointments, violence against women, and health problems in children were driven by the socio-economic characteristics of the population, most often to the disadvantage of social groups exposed to poor living conditions such as people reporting financial difficulties related to the health crisis.

### 4.1. External Validity

In France, the lockdown considerably altered the living and working conditions of the population for a period of almost two months [6]. We sought to determine whether the mental health of Reunionese adults was impacted by these changes depending on their socio-economic characteristics. A high level of stress during the lockdown was only explained by individual or personal characteristics: sex, age, financial difficulties related to the health crisis, and a history of chronic disease. In addition, a deteriorated psychological state was also explained by the neighborhood deprivation level depending on the condition of housing overcrowding. Of note, the latter finding was supported by the lockdown positive appreciation score analysis in the overall population. In the CoviPrev study aimed at describing the evolution of mental health and behaviors of the mainland French population during the COVID-19 pandemic, anxiety disorders occurring in the first two weeks of the lockdown were associated with socio-economic characteristics such as being a woman and reporting financial difficulties [12]. Interestingly, in this study, unlike what was observed in the entire sample (N = 4000), the people who reported financial difficulties did not have a reduction in anxiety prevalence between the two weeks of the survey, which supported, according to the authors, the reinforcement of SIH by the ongoing lockdown. Furthermore, the authors did not find any significant association between anxiety and the geographic area of residence [12]. In the same way, another work did not report a more significant association between depressive syndrome and the neighborhood economic status after adjustment for the type of housing and individual characteristics [19]. On Reunion Island, the characteristics of the microsocial environment are major determinants of health behaviors [20]. Several studies have confirmed the gender health inequality, exhibiting a higher prevalence of poor mental health outcomes among women compared to men in the first weeks of the initial lockdown [3,12,19,21]. We believe the gender inequalities present in society prior to the lockdown and those linked to the distribution of domestic tasks and childcare may have increased women’s mental load in the context of restriction to the home. In many international studies (for example: [22]), being younger increased the risk of adverse mental health outcomes, whereas our survey identified the 60–74-years-old age group as protective for mental health. With regards to the Reunionese historical context, the latter result could be further explained by the resilience of seniors who experienced poor living conditions during the decades following departmentalization in 1946 [9]. Consistent with our findings, a study conducted on mainland France during the first lockdown and dedicated to the specific investigation of people with chronic disease or disability (N = 1115) highlighted their psychological vulnerability [23].

The lockdown combined with stressful media coverage of the pandemic is likely to have caused an increase in “refuge behaviors” such as the use of screens, alcohol, tobacco, and psychotropic drugs. We sought to determine the socio-economic characteristics of Reunionese adults who showed an increase in addictive behaviors during the lockdown. To the best of our knowledge on the literature on the COVID-19 lockdown health impact, such addictive behaviors have not yet been analyzed using a count score, as we did under the assumption of a cumulative effect, but were considered one by one separately. For example, in a random subset (N = 14,237) of the EpiCov French cohort study, which includes Reunion Island, the daily alcohol consumption decreased from 2019 to 2020 in the ≥75-years-old age group [19]. Despite methodological discrepancies, this relationship was consistent with our finding showing a lower prevalence of increased addictive behavior in the ≥60-years-old age group compared to the 18–39-years-old age group. 

Limited access to essential stores had the effect of greatly reducing households’ food supply during the lockdown. Accordingly, we sought to identify the socio-economic profile of Reunionese adults who experienced the most difficulties in obtaining essential food items. Based on our data, this profile included financial difficulties related to the health crisis, age 18–39 years, and a potentially high mental load. In the CoviPrev study (N = 2000), a positive association was found between the financial situation perceived as difficult and weight gain upon the first lockdown [24]; weight gain was also associated with a reduction in fruit and vegetable consumption, which underlines the difficulty for the low-income population in accessing healthy food during the lockdown. In the Ré-Conf-ISS study, the participants aged 18–39 could have, on average, higher energy intake and food needs than those aged 75–90 due to a higher basal metabolic rate [25]. To some extent, this may explain the increased difficulties in food access for the youngest participants compared to the oldest. Equally, participants who came from a household with at least two children under 18 (and so being identified as exposed to a potentially high mental load) could have higher needs in food and higher difficulties in supplying the entire family compared to participants living alone or with only one child. 

Furthermore, for the diminution in physical activity, we found a socio-economic gradient among participants with optimal access to the outside during the lockdown. Thus, the probability of experiencing diminished physical activity during the lockdown decreased as the level of deprivation of large neighborhoods increased. This trend was consistent with that observed in the CoviPrev study, which used individual socio-economic indicators (instead of an ecological indicator such as ours). Thus, the reduction in physical activity was associated with an upper socio-professional category in men and a university degree in women, these data both being indicative of a high social status [26]. 

The lockdown and the fear of leaving one’s home due to the pandemic is likely to have resulted in medical follow-up reluctance. We sought to identify the socio-economic factors associated with changes in healthcare access and healthcare use in this context of isolation. Of note, the associations were all adjusted for the individual health state (i.e., history of chronic disease or health problem requiring regular visits to the doctor), which was classified as a major need factor in behavioral models for the use of health services [27]. According to the patient’s statement, it seems that during lockdown the local health system prioritized healthcare and medical follow-ups for the most impoverished Reunionese people (i.e., those identified as having no university degree and lacking complementary health insurance or benefiting from the free public complementary health insurance). However, for the sub-population of patients with chronic disease, they were more likely to have postponed medical appointments at the request of the medical secretariat. This relationship should be interpreted in the light of the decrease during the first French lockdown in both the activity of general practitioners and the number of hospitalizations outside of COVID-19 due to deprogramming and the application of sanitary measures [28,29]. 

Domestic violence including intimate partner violence was a hidden threat during the extended lockdown [30]. In particular, the issue of sheltering conditions and unsafe housing on the reinforcement of SIH has been raised [31]. In line with this, we found that, all things being equal, economic or psychological violence against women was more prevalent in the case of daily annoyances in the home and a high mental load. Unemployment as a proxy of economic instability [31] and a history of chronic disease were two additional vulnerability factors in the context of the low socio-economic setting of the Ré-Conf-ISS study. In Reunion Island, violence against women is a public health concern that has been little investigated and remains poorly understood [32].

Children’s health during the lockdown was another major issue. Our data suggest the role played by poor housing conditions and low incomes (and a broad lack of financial resources) in the context of lockdown and very limited work. The mediation of the effects of the first confinement (spring 2020) on child health by socio-environmental determinants has also been documented across studies conducted in mainland France. For example, for the children aged 0–2 years old, a retrospective survey conducted in the maternal and child protection service and in the nurseries of the city of Paris (N = 3685) showed that the frequencies of sleep disorders and relational difficulties were higher when lockdown took place in social housing than in individual housing [33]. Moreover, for the 8–9-years-old age group, the SAPRIS study (N = 4877) found that sleep disorders and socio-emotional difficulties during lockdown were more frequent among children from low-income households than among those from affluent households [34]. Finally, for the 9–18-years-old age group, the Confeado study (N = 3898) identified both poor housing and socio-economic conditions as factors associated with children’s psychological distress [35].

In summation, the observed SIH during the first lockdown in the French population could be explained by at least four mechanisms: first, the increased vulnerability of dependent populations such as children under 18 or people with comorbidities requiring regular care [36,37]; second, the growing income gap between lines of business (private/public) and socio-professional categories (e.g., the self-employed who found themselves without work and income, workers who became partially unemployed, and managers and intellectual professionals who were able to maintain their income by working at home) [38]; third, the accumulation of exposures to health risks (e.g., poor housing and living conditions in a precarious neighborhood); fourth, the reinforcement of power relations based on gender, geographical origin, or social class [39].

### 4.2. Internal Validity

The Ré-Conf-ISS telephone survey, which was designed and conducted rapidly to address the emergency health situation in Reunion Island, has several limitations. First, the absence of random sampling means that our sample is not representative of the adult Reunionese population in 2020, notably in terms of sex, age, housing occupancy status, usually living alone, and complementary health insurance status. However, controlling for sex, age, and a mix of individual, collective, and ecological socio-economic conditions in the fully adjusted multivariable regression models may have reduced the impact of selection bias on our main findings. Second, households placed on a red list or lacking a fixed telephone subscription were not represented in the Ré-Conf-ISS statistical sample, nor were members of the most vulnerable and precarious social groups (e.g., people living in hostels, institutions, or on the street). Under these conditions of analysis, the SIH identified in our survey were measured based on a socio-economic gradient that was probably less contrasted than the one observed in reality. In other words, our results are likely to be conservative. Third, the study questionnaire lacked a validated coping scale [40]. Nevertheless, it included a score based on relevant local items describing lockdown positive appreciation. Fourth, as with many other studies dealing with the lockdown population health impact [3,34,39,40], our data were self-reported, which means that the health states of respondents were not confirmed by research examination or medical claims data. The health data may therefore be undermined by reporting bias such as a possible lack of awareness of the child’s health problems by the father compared to the mother. However, the impact of this classification bias was likely limited by the use of several health questions from an interdisciplinary and multi-partner research project on the same topic [41] and by our reliance on a survey platform that has, for many years, been in charge of the conduct of telephone interviews in local population-based epidemiological studies [42,43,44].

The Ré-Conf-ISS telephone survey also has several strengths. First, it was a cross-sectional survey with retrospective data collection on the lockdown period showing a short delay from the end of this landmark event (min–max: 2 days–2.5 months). Thus, recall (memory) bias on behavior, health, or living conditions during the lockdown was likely low. Second, the survey was implemented during a 2.5-month period (May–July, 2020) when the local SARS-CoV-2 epidemic proved, a posteriori, to still be underdeveloped (regional cumulative incidence < 1‰ [45]). Therefore, our findings on negative mental health were probably not explained by direct exposure to the epidemic virus but by the indirect effect of the pandemic or lockdown. Third, participants were selected from the 114 large neighborhoods of Reunion Island all being classified according to the Insee typology [10], which is based on several social determinants of health [13] (e.g., economic insecurity, living standard, and family structure) involved in the SIH indicators [46]. In addition, controlling housing conditions during the lockdown as exposure factors or effect modifier factors may have improved the lockdown-related SIH measure.

### 4.3. Implications for Research and Public Health

Along with other research initiatives, our study responds to the need expressed by the international scientific community for data on the economic, social, psychosocial, and health consequences of the universal prevention measures implemented to tackle the SARS-CoV-2 epidemic. The findings support that several behaviors and health states related to the eight-week lockdown period were driven by the socio-economic characteristics of the population. In accordance with the study’s conceptual model, most of the fully adjusted multivariable models highlighted the cumulative effect of the social conditions experienced during the lockdown on individual health. Some determinants operated at a level of individual exposure, such as certain socio-economic or medical characteristics, while other determinants came under the socio-environmental sphere, such as housing conditions and the level of deprivation in the neighborhood of residence. Despite the limits inherent in cross-sectional surveys for the causality inference [47], this work helps to understand the possible articulation between certain economic and social dimensions and the development of local SIH in the context of societal lockdown. Public health researchers in interaction with social actors on Reunion Island could elaborate on these findings to propose adjustments and supportive mechanisms for the prevention of health problems related to long-term complete lockdowns, keeping in mind the need for reducing SIH in the Reunionese population.

Last but not least, the focus on contextual, collective, and individual health determinants of a specific territory is relevant for population health intervention research. In this way, understanding why a given population is concerned by disparities in health according to the socio-economic position or living conditions could help to build programmatic and policy interventions that are fair for everyone [48].

## 5. Conclusions

In spring 2020, Reunion Island was locked down before being affected by the SARS-CoV-2 epidemic. The pre-existing socio-economic vulnerabilities of Reunionese people led to significant health inequalities in terms of broad wellbeing outcomes. Consequently, the lockdown public health strategy was completely ineffective at curtailing the epidemic since it had not yet impacted the island.

## Figures and Tables

**Figure 1 ijerph-19-13932-f001:**
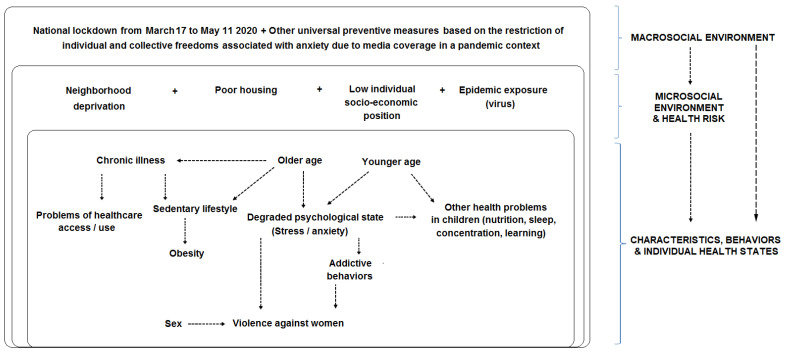
Conceptual model and socio-ecological framework.

**Table 1 ijerph-19-13932-t001:** Exposure factors, demographic factors, baseline health state in the Ré-Conf-ISS study population and regional representativeness.

Characteristics	Ré-Conf-ISS Study (N = 892)		Reunion Island
	nmiss	*n* (%)	(%)
**Large neighborhood deprivation level**: ^1^			
Group 1 (most deprived)	0	100 (11.2)	(11.3) ^2^
Group 2	0	202 (22.6)	(22.6) ^2^
Group 3	0	173 (19.4)	(19.4) ^2^
Group 4	0	262 (29.4)	(29.4) ^2^
Group 5 (less deprived)	0	155 (17.4)	(17.3) ^2^
**Housing conditions during the lockdown**:			
Daily annoyances in the home (Yes)	0	184 (20.6)	-
Poor Internet access (Yes)	4	407 (45.8)	-
A potentially high mental load (Yes)	2	214 (24.0)	-
Living in overcrowded conditions (Yes)	35	275 (32.1)	-
Optimal access to the outside (Yes)	8	216 (24.4)	-
**Individual socio-economic position**:			
Experience of financial difficulties ^3^ (Yes)	3	279 (31.4)	-
University degree as education level (Yes)	16	187 (21.3)	(20.2) ^4^
Housing occupancy status (Homeowners)	5	493 (55.6)	(50.7) ^4^
Housing occupancy status (Renters)	5	280 (31.5)	(45.2) ^4^
Housing occupancy status (Rent-free)	5	114 (12.9)	(4.1) ^4^
Usually living alone (Yes)	0	163 (18.3)	(11.1) ^4^
Socio-professional category (Employed)	13	388 (44.1)	-
Socio-professional category (Unemployed)	13	252 (28.7)	-
Socio-professional category (Retirees)	13	239 (27.2)	-
Complementary health insurance status (Negative ^5^)	9	243 (27.5)	(43.5) ^6^
**SARS-CoV-2 epidemic exposure:**			
Self-reported infection (Yes ^7^)	0	2 (0.2)	-
Knowledge of an infected close relative ^8^ (Yes)	2	48 (5.4)	-
**Demographic factors and baseline health state:**			
Sex (Men)	0	321 (36.0)	(46.2) ^9^
Age (18–39 yrs)	2	239 (26.9)	(33.9) ^9^
Age (40–59 yrs)	2	371 (41.7)	(39.7) ^9^
Age (60–74 yrs)	2	195 (21.9)	(19.1) ^9^
Age (75–90 yrs)	2	85 (9.6)	(7.3) ^9^
History of chronic disease ^10^ (Yes)	0	342 (38.3)	-

^1^ Group 1 consisted of “urban neighborhoods facing multiple socio-economic difficulties”; Group 2 consisted of “predominantly rural neighborhoods inhabited by poor homeowners”; Group 3 consisted of “vulnerable neighborhoods located close to city centers”; Group 4 consisted of “less poor neighborhoods located far from city centers”; Group 5 consisted of “better off neighborhoods” [10]. ^2^ Percentages taken from the population ≥ 18 years ([10]); ^3^ related to the COVID-19 health crisis; ^4^ 2018 census data (Insee); ^5^ lacking complementary health insurance or benefiting from the free public complementary health insurance; ^6^ 2014 survey data [18]; ^7^ with PCR test confirmation; ^8^ living on Reunion Island or elsewhere; ^9^ 2020 census data (Insee) for ≥18 years; ^10^ health problem requiring regular visits to the doctor. nmiss: number of observations with missing data. *n*: number of observations. %: percentage. First column, in bold: the main categories of factors.

**Table 2 ijerph-19-13932-t002:** Endpoints in the Ré-Conf-ISS study population (N = 892).

Behaviors and Individual Health States Related to the COVID-19 Lockdown Period	nmiss	*n* (%) or Median (Min–Max)
High level of stress (Yes)	5	179 (20.2)
Deteriorated psychological state (Yes)	0	422 (47.3)
Lockdown positive appreciation (No or don’t know)	1	487 (54.7)
Number of relaxing activities engaged in	14	0 (0–13)
Increase in ≥1 addictive behavior (Yes)	0	516 (57.8)
Number of difficulties in accessing food	3	1 (0–5)
Difficulty in accessing food (No)	0	389 (43.6)
Decrease in physical activity (Yes)	0	553 (62.0)
Delayed medical appointment (At the request of the secretariat)	4	191 (21.5)
Delayed medical appointment (On its own initiative)	4	148 (16.7)
Women (*n* = 571) suffering at least one situation of violence (Yes)	0	22 (3.9)
Number of health problems in children	3	1 (0–5)
At least one health problem in children (Yes)	0	242 (68.4)

Health problems in children was based on the statements by 354 parents confined with their child(ren) during the lockdown. nmiss: number of observations with missing data. *n*: number of observations. %: percentage. min–max: range.

**Table 3 ijerph-19-13932-t003:** Factors independently associated with negative mental health endpoints in the adult population during the COVID-19 lockdown (Ré-Conf-ISS cross-sectional study: Reunion Island, 13 May and 22 July 2020)—fully adjusted multivariable models (M4).

Factors	High Level of Stress ^1^		Deteriorated Psychological State ^2^			
			Living in Overcrowded Conditions:			
			Yes		No	
	aOR (95% CI)	*p*	aOR (95% CI)	*p*	aOR (95% CI)	*p*
**Increase in large neighborhood deprivation level** (continuous)	1.03 (0.88–1.22)	0.698	0.86 (0.71–1.04)	0.127	1.17 (1.01–1.37)	0.046
**Housing conditions during the lockdown:**						
Daily annoyances in the home (Yes vs. No)	1.13 (0.74–1.73)	0.580	-	-	-	-
**SARS-CoV-2 epidemic exposure:**						
Knowledge of an infected close relative ^3^ (Yes vs. No/don’t know)	1.36 (0.72–2.57)	0.341	1.79 (0.57–5.64)	0.323	1.06 (0.46–2.46)	0.889
**Individual socio-economic position:**						
Experience of financial difficulties ^4^ (Yes vs. No/don’t know)	1.59 (1.04–2.41)	0.031	2.36 (1.41–3.96)	0.001	1.99 (1.28–3.10)	0.002
**Demographic factors and baseline health state:**						
Sex (Man vs. Woman)	0.48 (0.32–0.72)	<0.001	0.87 (0.54–1.38)	0.548	0.85 (0.62–1.18)	0.335
Age (60–74 yrs vs. 18–59 yrs)	0.48 (0.29–0.79)	0.004	0.73 (0.27–2.00)	0.543	0.55 (0.35–0.87)	0.010
Age (75–90 yrs vs. 18–59 yrs)	0.83 (0.46–1.49)	0.529	1.47 (0.20–10.99)	0.709	0.69 (0.39–1.21)	0.199
History of chronic disease ^5^ (Yes vs. No/don’t know)	1.63 (1.13–2.35)	0.009	2.20 (1.26–3.83)	0.005	1.09 (0.74–1.59)	0.666

^1^ A high level of stress during the lockdown was a binary endpoint defined by a score > 6 on a scale ranging from 0 (no stress) to 10 (maximum imaginable stress) [15]. The reference category was “No high level of stress” (score ≤ 6). ^2^ A deteriorated psychological state during the lockdown was a binary endpoint defined by the presence of at least one of the following five situations: currently not feeling calm in daily life and/or lockdown-related worsening of a psychological condition pre-existing the epidemic and/or a psychological condition triggered by the epidemic or lockdown and/or perception of one’s social relationships as very bad or somewhat bad and/or worries about ones’ job or work situation in the immediate future. The reference category is “No deteriorated psychological state”. ^3^ Living on Reunion Island or elsewhere. ^4^ Related to the COVID-19 health crisis. ^5^ Health problem requiring regular visits to the doctor. Figures are from multivariable binary logistic regression models adjusted for large neighborhood deprivation level, housing conditions, indirect epidemic exposure, individual socio-economic position, demographic factors, and baseline health state. For the deteriorated psychological state endpoint, multivariable binary logistic regression model was stratified by living in overcrowded conditions during the lockdown (Yes/No). aOR: adjusted odds ratio. 95% CI: 95% confidence interval. *p*: *p*-value. vs.: versus. First column, in bold: the main categories of factors.

**Table 4 ijerph-19-13932-t004:** Factors independently associated with lockdown positive appreciation and addictive behaviors’ endpoints in the adult population during the COVID-19 lockdown (Ré-Conf-ISS cross-sectional study: Reunion Island, 13 May and 22 July 2020)—fully adjusted multivariable models (M4).

Factors	Lockdown Positive Appreciation ^1^				Increase in ≥1 Addictive Behavior ^2^	
	Level of Engagement in Relaxing Activities		No Positive Appreciation or Don’t Know			
	aIRR (95% CI)	*p*	aOR (95% CI)	*p*	aOR (95% CI)	*p*
**Increase in large neighborhood deprivation level** (continuous)	0.99 (0.97–1.02)	0.615	1.24 (1.07–1.42)	0.003	0.89 (0.80–0.99)	0.026
**Housing conditions during the lockdown:**						
Poor Internet access (No vs. Yes)	-	-	-	-	1.20 (0.89–1.62)	0.231
A potentially high mental load (Yes vs. No)	1.01 (0.96–1.07)	0.628	0.71 (0.51–0.98)	0.040	-	-
Optimal access to the outside (Yes vs. No)	1.10 (1.03–1.17)	0.003	0.94 (0.69–1.27)	0.672	-	-
**Individual socio-economic position:**						
University degree as education level (Yes vs. No)	1.11 (1.05–1.18)	0.001	0.82 (0.57–1.16)	0.263	1.86 (1.26–2.76)	0.002
Housing occupancy status (Homeowners vs. Rent-free)	1.02 (0.98–1.06)	0.352	0.96 (0.74–1.26)	0.780	-	-
Housing occupancy status (Renters vs. Rent-free)	0.96 (0.93–0.99)	0.008	1.05 (0.81–1.36)	0.712	-	-
Housing occupancy status (Renters vs. Homeowners)	-	-	-	-	1.84 (1.29–2.61)	<0.001
Housing occupancy status (Rent-free vs. Homeowners)	-	-	-	-	1.20 (0.82–1.74)	0.345
**Demographic factors and baseline health state:**						
Sex (Man vs. Woman)	0.97 (0.92–1.03)	0.349	1.62 (1.21–2.16)	0.001	0.85 (0.63–1.13)	0.255
Age (40–59 yrs vs. 18–39 yrs)	1.00 (0.96–1.03)	0.790	0.82 (0.65–1.02)	0.077	0.95 (0.66–1.36)	0.787
Age (60–74 yrs vs. 18–39 yrs)	0.96 (0.91–1.02)	0.165	1.06 (0.78–1.44)	0.712	0.48 (0.32–0.70)	<0.001
Age (75–90 yrs vs. 18–39 yrs)	1.01 (0.94–1.07)	0.880	1.87 (1.22–2.87)	0.004	0.45 (0.25–0.83)	0.010
History of chronic disease ^3^ (Yes vs. No/don’t know)	0.98 (0.92–1.03)	0.384	1.12 (0.85–1.48)	0.414	0.78 (0.56–1.09)	0.140

^1^ The lockdown positive appreciation score was a count endpoint defined as the sum of behaviors or attitudes for having engaged in at least one of the following thirteen relaxing activities: refocusing, enjoying the quiet, breathing less polluted air, having conversations, reading, writing, watching movies/series, playing games, cooking, DIY, gardening, doing nothing, doing something else that feels good. The higher the score, the better the positive appreciation of the lockdown. A zero (minimum) value meant no positive appreciation of the lockdown or don’t know. ^2^ An increase in addictive behaviors during the lockdown was a binary endpoint. It concerned the use of at least one of the following items: screens, alcohol, tobacco (cigarette/e-cigarette), or psychotropic drugs (natural or synthetic cannabinoid-based drugs), and participation in virtual happy hours. The reference category was “No increase in addictive behavior during the lockdown”. ^3^ Health problem requiring regular visits to the doctor. Figures are from multivariable zero-inflated negative binomial (ZINB) regression model (for the lockdown positive appreciation score) or binary logistic regression model (for an increase in at least one addictive behavior during the lockdown) adjusted for large neighborhood deprivation level, housing conditions, individual socio-economic position, demographic factors, and baseline health state. In the zero-inflated part of the ZINB regression model, the reference category was “A positive appreciation of the lockdown for at least one of the thirteen relaxing activities”. aIRR: adjusted incidence rate ratio. aOR: adjusted odds ratio. 95% CI: 95% confidence interval. *p*: *p*-value. vs.: versus. First column, in bold: the main categories of factors.

**Table 5 ijerph-19-13932-t005:** Factors independently associated with nutrition endpoints in the adult population during the COVID-19 lockdown (Ré-Conf-ISS cross-sectional study: Reunion Island, 13 May and 22 July 2020)—fully adjusted multivariable models (M4).

Factors	Difficulties in Accessing Food ^1^				Decrease in Physical Activity ^2^			
					Optimal Access to the Outside:			
	Number of Difficulties		No Difficulty		Yes		No	
	aIRR (95% CI)	*p*	aOR (95% CI)	*p*	aOR (95% CI)	*p*	aOR (95% CI)	*p*
**Increase in large neighborhood deprivation level** (continuous)	0.98 (0.94–1.03)	0.497	1.10 (0.94–1.28)	0.248	0.69 (0.54–0.88)	0.003	1.02 (0.88–1.18)	0.808
**Housing conditions during the lockdown:**								
A potentially high mental load (Yes vs. No)	1.05 (0.91–1.21)	0.482	0.54 (0.30–0.97)	0.041	-	-	-	-
Living in overcrowded conditions (Yes vs. No)	1.13 (0.99–1.30)	0.071	1.45 (0.86–2.46)	0.162	-	-	-	-
Poor Internet access (No vs. Yes)	-	-	-	-	1.05 (0.49–2.28)	0.897	1.26 (0.92–1.73)	0.148
**Individual socio-economic position:**								
University degree as education level (Yes vs. No)	-	-	-	-	1.03 (0.46–2.28)	0.948	1.79 (1.17–2.75)	0.008
Usually living alone (Yes/No)	0.87 (0.73–1.04)	0.123	1.68 (0.97–2.92)	0.065	-	-	-	-
Experience of financial difficulties ^3^ (Yes vs. No/don’t know)	1.21 (1.06–1.38)	0.004	0.99 (0.62–1.57)	0.960	-	-	-	-
**Demographic factors and baseline health state:**								
Sex (Man vs. Woman)	0.94 (0.83–1.07)	0.338	1.10 (0.71–1.70)	0.670	0.64 (0.31–1.32)	0.230	1.12 (0.83–1.51)	0.476
Age (40–59 yrs vs. 18–39 yrs)	1.11 (0.98–1.26)	0.092	0.73 (0.51–1.06)	0.095	0.71 (0.36–1.42)	0.336	1.19 (0.77–1.82)	0.434
Age (60–74 yrs vs. 18–39 yrs)	1.07 (0.90–1.28)	0.416	1.16 (0.78–1.72)	0.456	1.09 (0.49–2.45)	0.827	1.55 (0.96–2.52)	0.074
Age (75–90 yrs vs. 18–39 yrs)	0.85 (0.63–1.15)	0.286	2.87 (1.56–5.27)	0.001	0.47 (0.15–1.40)	0.174	0.57 (0.30–1.12)	0.102
History of chronic disease ^4^ (Yes vs. No/don’t know)	0.97 (0.86–1.09)	0.578	1.02 (0.68–1.52)	0.932	0.56 (0.29–1.08)	0.085	0.96 (0.65–1.42)	0.838

^1^ The score of difficulties in accessing food during the lockdown was a count endpoint defined as the sum of difficulties for the following items: fresh fruits and vegetables, garlic/onions, eggs, flour, and other essential food items. The higher the score, the higher the number of difficulties. A zero (minimum) value meant no difficulty. ^2^ A decrease in physical activity during the lockdown was a binary endpoint defined by the presence of at least one of the following three situations: sports practice at a club or an association before the lockdown and/or use of a neighborhood sports facility before the lockdown and/or perceived decrease in physical activity during the lockdown. The reference category was “No increase in physical activity during the lockdown”. ^3^ Related to the COVID-19 health crisis. ^4^ Health problem requiring regular visits to the doctor. Figures are from multivariable zero-inflated negative binomial (ZINB) regression model (for the score of difficulties in accessing food during the lockdown) or binary logistic regression model (for an increase in physical activity during the lockdown) adjusted for large neighborhood deprivation level, housing conditions, individual socio-economic position, demographic factors, and baseline health state. In the zero-inflated part of the ZINB regression model, the reference category was “≥ 1 difficulty in accessing food during the lockdown”. For decrease in physical activity during the lockdown, multivariable binary logistic regression model was stratified by optimal access to the outside during the lockdown (Yes/No). aIRR: adjusted incidence rate ratio. aOR: adjusted odds ratio. 95% CI: 95% confidence interval. *p*: *p*-value. vs.: versus. First column, in bold: the main categories of factors.

**Table 6 ijerph-19-13932-t006:** Factors independently associated with health care use endpoint in the adult population during the COVID-19 lockdown (Ré-Conf-ISS cross-sectional study: Reunion Island, 13 May and 22 July 2020)—fully adjusted multivariable model (M4).

Factors	Delayed Medical Appointment at the Request of the Medical Secretariat ^1^	
	aOR (95% CI)	*p*
**Increase in large neighborhood deprivation level** (continuous)	1.07 (0.91–1.27)	0.406
**Housing conditions during the lockdown:**		
A potentially high mental load (Yes vs. No)	1.36 (0.86–2.14)	0.183
Optimal access to the outside (Yes vs. No)	0.86 (0.56–1.32)	0.501
Daily annoyances in the home (Yes vs. No)	1.46 (0.95–2.26)	0.086
**Individual socio-economic position:**		
University degree as education level (Yes vs. No)	1.58 (1.08–2.30)	0.018
Complementary health insurance status (Positive vs. Negative) ^2^	1.62 (1.05–2.51)	0.030
**Demographic factors and baseline health state:**		
Sex (Man vs. Woman)	0.73 (0.52–1.03)	0.074
Age (40–90 yrs vs. 18–39 yrs)	0.72 (0.46–1.12)	0.150
History of chronic disease ^3^ (Yes vs. No/don’t know)	2.25 (1.46–3.48)	<0.001

^1^ A medical appointment during the lockdown delayed at the request of the medical secretariat was a binary endpoint. The reference category was “No delayed medical appointment at all or medical appointment delayed for another reason (than at the request of the medical secretariat and on its own initiative)”. ^2^ Negative: lacking complementary health insurance or benefiting from the free public complementary health insurance. Positive: benefiting from a private complementary health insurance. ^3^ Health problem requiring regular visits to the doctor. Figures are from a multivariable binary logistic regression model adjusted on large neighborhood deprivation level, housing conditions, individual socio-economic position, demographic factors, and baseline health state. aOR: adjusted odds ratio. 95% CI: 95% confidence interval. *p*: *p*-value. vs.: versus. First column, in bold: the main categories of factors.

**Table 7 ijerph-19-13932-t007:** Factors independently associated with economic or psychological violence against women and health problems in children during the COVID-19 lockdown (Ré-Conf-ISS cross-sectional study: Reunion Island, 13 May and 22 July 2020)—fully adjusted multivariable models (M4).

Factors	Women Suffering at Least One Situation of Violence ^1^		Number of Health Problems in Children (<18 yrs) ^2^		At Least One Health Problem in Children (<18 yrs) ^3^	
	aOR (95% CI)	*p*	aIRR (95% CI)	*p*	aOR (95% CI)	*p*
**Increase in large neighborhood deprivation level** (continuous)	0.95 (0.67–1.34)	0.765	0.96 (0.87–1.06)	0.397	0.90 (0.72–1.12)	0.348
**Housing conditions during the lockdown:**						
Daily annoyances in the home (Yes vs. No)	2.39 (1.07–5.34)	0.034	1.19 (0.94–1.51)	0.140	2.09 (1.14–3.85)	0.017
A potentially high mental load (Yes vs. No)	3.33 (1.38–8.05)	0.007	-	-	-	-
**Individual socio-economic position:**						
Parent usually living alone (Yes/No)	-	-	1.56 (0.99–2.46)	0.052	1.59 (0.38–6.57)	0.525
Experience of financial difficulties ^4^ (Yes vs. No/don’t know)	1.03 (0.44–2.40)	0.944	1.42 (1.14–1.77)	0.002	1.49 (0.90–2.47)	0.125
Socio-professional category (Unemployed vs. Employed)	3.02 (1.05–8.71)	0.040	-	-	-	-
Socio-professional category (Retirees vs. Employed)	0.31 (0.04–2.24)	0.247	-	-	-	-
Housing occupancy status (Renters vs. Homeowners)	1.59 (0.63–4.02)	0.328	-	-	-	-
Housing occupancy status (Rent-free vs. Homeowners)	1.72 (0.53–5.57)	0.364	-	-	-	-
**Demographic factors and baseline health state:**						
Sex of parent (Man vs. Woman)	-	-	0.76 (0.58–0.99)	0.046	0.59 (0.36–0.94)	0.026
Age of parent (40–90 yrs vs. 18–39 yrs)	-	-	0.94 (0.76–1.15)	0.550	0.69 (0.44–1.08)	0.103
History of chronic disease ^5^ (Yes vs. No/don’t know)	5.22 (2.25–12.09)	<0.001	-	-	-	-

^1^ Violence against women during the lockdown was defined by nine situations of economic or psychological abuse linked to the lockdown and the COVID-19 pandemic. Women suffering at least one situation of violence was a binary endpoint. The reference category was “No violence suffered during the lockdown”. ^2^ Health problems in children during the lockdown concerned the entire sibling group and consisted of the following: worry, anxiety, stress, sleep and eating disorders, concentration or attention problems, and learning difficulties. They were identified based on the statements by 354 parents confined with their child(ren) during the lockdown and summarized in a sum score. The higher the score, the lower the health of children. ^3^ At least one health problem in children was a binary endpoint based on the health children score ≥ 1 (Yes/No). The reference category was “No health problem in children during the lockdown”. ^4^ Related to the COVID-19 health crisis. ^5^ Health problem requiring regular visits to the doctor. Figures are from multivariable binary logistic regression model (for women and children binary endpoints) or negative binomial regression model with an offset [=log(size of entire sibling group)] to adjust the at-risk population size (for the health children score) adjusted for large neighborhood deprivation level, housing conditions, individual socio-economic position, demographic factors (except for economic or psychological violence against women), and baseline health state (except for health children endpoints). aIRR: adjusted incidence rate ratio. aOR: adjusted odds ratio. 95% CI: 95% confidence interval. *p*: *p*-value. vs.: versus. First column, in bold: the main categories of factors.

## Data Availability

The data presented in this study are not publicly available due to restrictions from the study sponsor (CHU Réunion). Any request for the transmission of data will be subject to an agreement between the applicant and the CHU Réunion (data transfer agreement).

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
