# Peer review of "Health Impacts of the COVID-19 Lockdown Measure in a Low Socio-Economic Setting: A Cross-Sectional Study on Reunion Island"

_ijerph, 2022, doi:10.3390/ijerph192113932_

Round 1

Reviewer 1 Report

The authors have presented compelling reasons for exploring a critical topic, and the study appears to have reasonable scientific soundness. However, the paper can be improved in the following ways:

-       In line 49, "expected to have affected" should be changed to "affected".

-       In line 62, what is Insee? Please elaborate on this and provide a citation.

-       The sentence in line 64 makes some important points but needs to be backed up by the literature--please provide citations.

-       In the Methods section, it is unclear what the inclusion criteria for the quota sampling are –are these the five groups discussed later on? If so, please move that description to earlier in the Methods section when discussing the quota sampling.

-       The conceptual model is a nice reflection of the socio-ecological framework. This discussion should be moved up to the Introduction to provide the conceptual underpinnings of this study. In the model, however, it looks like only older age leads to stress/anxiety—what about other potential predictors? As is, this model is fairly narrow.

-       The paragraph in lines 179-236 is extremely long and difficult to follow. Please consider using indented subheadings and bullet points if necessary.

-       In lines 254-277, please simplify the discussion of the stepwise regression analysis. As written, it is too long and difficult to follow.

-       Figure 2 is not needed in the main text and can be moved to an Appendix.

-       The manuscript uses the terms “endpoints” often. What is meant by this term? Do the authors mean outcome variables? If so, please state this.

-       In terms of the results, this manuscript, in general, presents too many findings, and hence, there are also too many tables and results to discuss in the Results and Discussion sections. While all of the information is important, it is too much information for the reader to absorb at one time. The authors should consider prioritizing the most important/significant findings and perhaps also present remaining findings in another paper. This will allow for more focused papers that are conceptually driven and adequately discussed.

-       As suggested above, the Discussion section needs to be cut down once the Results section is reduced.

-       The implications section needs to be moved from the Conclusion to the Discussion, and it should also be expanded upon. Currently, the Discussion has too much emphasis on the findings and how they fit into the literature (this can be summarized a bit), and it does not adequately discuss the importance of these findings for programmatic and policy interventions or future research directions.

Reviewer 2 Report

Despite being geographically very restricted, the article is interesting, clear and reveals scientific rigor.

As improvements we propose that:

1 - The sample, and the respective characterization of the individuals, should appear before the Results and in a separate section.

2 - Page 7 – in figure 2, should be stated what means unavailables; refusals; ineligibles, because it is not clear what characterizes each of these groups. For example, what were the reasons for being considered ineligible?

3 - The justification for the relevance of this study for the future must be reinforced, taking into account that the pandemic is practically over and it is not likely that we will go through other lockdowns soon.

Reviewer 3 Report

Thanks for the opportunity to review this interesting and significant paper.

The section of Literature review is well written. It addresses the issue in various perspectives and contains a sufficient number of relevant and important scientific studies. Also, you can add: https://doi.org/10.1007/s10198-022-01506-1, https://doi.org/10.1177/02654. However, the hypotheses miss. They have to be included at the end of literature review part. I also, appreciate if you can present all variables in tabular form and I think important to present the final form of questionnaires in appendix.

Also, please insist on the international dimension of the findings of this research: to what extent are these findings valuable within the worldwide context?

Round 2

Reviewer 1 Report

Extensive proofreading of the entire manuscript is still needed as many grammatical errors are present. 

Reviewer 3 Report

I have read the entire manuscript carefully and I have not any comments for its authors, because they have addressed all raised issues. However, there is a minor flaw in this article which need to be addressed. My suggestion is that the authors mention another important limitation of the study, the lack of coping scale. After that, I consider this paper worth publishing in IJERPH .
